# Covalent Organic Framework (COF): A Drug and Carrier to Attenuate Retinal Ganglion Cells Death in an Acute Glaucoma Mouse Model

**DOI:** 10.3390/polym14163265

**Published:** 2022-08-10

**Authors:** Ke Yao, Xin Liang, Guiyang Zhang, Yan Rong, Qiuxiang Zhang, Qiaobo Liao, Hong Zhang, Kai Xi, Junming Wang

**Affiliations:** 1Department of Ophthalmology, Tongji Hospital, Tongji Medical College of Huazhong University of Science and Technology, Wuhan 430000, China; 2School of Chemistry and Chemical Engineering, Nanjing University, Nanjing 210000, China

**Keywords:** covalent organic framework, rapamycin, retinal ischemia–reperfusion, glaucoma

## Abstract

Purpose: We aim to investigate the use of covalent organic framework (COF) nanoparticles in the local treatment of glaucoma, both as a means of protecting retinal ganglion cells (RGCs), and as a carrier for delayed release of the medication rapamycin following a single intravitreal injection. Methods: a water-dispersible COF, and a COF-based nanoplatform for rapamycin release (COF-Rapa) was constructed. C57BL/6J mice were randomly divided into four groups: intravitreal injection of 1.5 µL normal saline (NS), COF (0.67 ng/µL), rapamycin (300 µM) or COF-Rapa (0.67 ng/µL-300 µM), respectively. The ischemia–reperfusion (I/R) model was established to mimic high intraocular pressure (IOP)-induced retinal injury in glaucoma. Labeling of RGCs by Fluoro-Gold and retinal electroretinogram were used to evaluate retinal function. Immunohistochemistry and Western blotting analyses of retinas were performed. Results: COF nanoparticles were delivered in vitro and in vivo. Six weeks after the COF injection, the number of RGCs was unaffected. In addition, the number of RBPMS-positive RGCs, GFAP-positive astrocytes and Iba1-positive microglia did not differ from the normal control. COF could effectively reduce RGCs death, improve phototransduction function and alleviate the overactivation of microglia compared to NS control after retinal I/R injury. Within six weeks, the mammalian target of rapamycin complex 1 (mTORC1) signaling pathway in the retinas could be inhibited by a single intravitreal injection of COF-Rapa. Compared with single COF administration, COF-Rapa significantly reduced the inflammatory reaction after retinal I/R injury. Conclusions: COF may act as both an RGC protection agent and a carrier for prolonged rapamycin release. This research may lead to the development of novel RGC protection agents and drug delivery techniques, as well as the creation of multifunctional COF-based biomaterials for glaucoma retinopathy.

## 1. Introduction

It is estimated that glaucoma will affect 111.8 million people by 2040 [1]. As a complex neurodegenerative disease, glaucoma is a worldwide leading cause of irreversible vision loss [2]. Previous studies have shown that the progressive loss of retinal ganglion cells (RGCs), and thinning of the retinal nerve fiber layer can lead to visual field loss [1,2]. In acute glaucoma, intraocular pressure (IOP) increases rapidly in a short time, in contrast to the slow progression of the disease in other types, leading to retinal ischemia and progressive RGC death [2], the exact mechanism is complicated and needs further research. Nowadays, researchers are trying to find suitable targets and drugs or carriers to prevent or attenuate the loss of RGCs in the pathological glaucoma.

As an emerging class of porous crystalline materials, covalent organic frameworks (COFs) have gained considerable interest due to their extensive applications in smart sensing, catalysis, energy, and separation science [3,4,5]. Recent studies have revealed that COFs can serve as drug-delivery carriers with high loading capacity and efficient drug-release behavior [6,7,8]. Zhang et al. have synthesized a water-dispersible polymer-COF nanocomposite (denoted as PEG-CCM@APTES-COF-1) via the self-assembly of polyethylene-glycol-modified monofunctional curcumin derivatives (PEG-CCM) and amine-functionalized COF-1 (PEG-CCM@APTES-COF-1, abbreviated as COF) [8]. This water-dispersible polymer-COF nanocomposite has been a smart drug delivery carrier with remarkable anticancer therapeutic efficiency [8]. However, the effect and application of COF in eyes has not been studied.

Rapamycin forms a gain-of-function complex with the 12 kDa FK506-binding protein, which binds and acts explicitly as an allosteric inhibitor of mammalian target of rapamycin complex 1 (mTORC1) [9,10]. Several studies have demonstrated the good protective effect of rapamycin for glaucoma. Su et al. demonstrated that rapamycin was neuroprotective in a rat model of chronic hypertensive glaucoma by inhibiting the release of neurotoxic mediators and/or directly suppressing RGCs apoptosis [11]. Using a mouse model of retinal ischemia, Rossella et al. demonstrated that rapamycin could induce autophagy in the insulted retina to promote RGC survival [12]. Harder et al. found that mTORC1 activation occurs early in glaucoma using DBA2J mice with spontaneous glaucoma, and rapamycin treatment protected them from glaucoma associated with decreased energy consumption [13]. However, all the researchers administered rapamycin daily intraperitoneally (i.p.) or via drinking water at a dose of 2–10 mg/kg/d, lasting from one week to six months. Long-term and high-dose systemic medication raises concerns about unknown effects of rapamycin application on other organs. The intravitreal injection is a promising route of administration of circumventing these obstacles, as it has an excellent local and systematic safety profile. However, a single intravitreal injection of rapamycin is effective only within a short period, as shown by our following experiments, while repeated injection can increase the risk of endophthalmitis, retinal detachment, and traumatic cataract, etc. [14,15]. Thus, finding a carrier that can hold and release rapamycin gradually is therefore crucial to rapamycin’s effectiveness and safety in the treatment of glaucoma retinopathy.

In this study, we described a sustained release system for rapamycin based on PEG-CCM@APTES-COF-1, which was an excellent approach to protecting RGCs in a mouse model of acute glaucoma. To the best of our knowledge, this study is the first to introduce the burgeoning field of polymer-COF assembly in the local management of glaucoma retinopathy.

## 2. Materials and Methods

### 2.1. Synthesis and Self-Assembly of APTES-COF-1@Rapamycin/Null and PEG-CCM

Based on the steps and methods described before [8], we develop a facile synthesis of a polymer-COF nanocomposite via the self-assembly of polyethylene-glycol-modified monofunctional curcumin derivatives (PEG-CCM) and amine-functionalized COF-1 (APTES-COF-1). The entire structure of nanocomposites can be regarded as a micelle with APTES-COF-1 as the oil phase and PEG-CCM as the surfactant. APTES-COF-1 was synthesized, and APTES-COF-1@Rapamycin/Null and PEG-CCM were assembled (abbreviated as COF-Rapa and COF separately, encapsulated with rapamycin or not). Inside the cell, PEG-CCM is unplugged, and rapamycin is released by COF-Rapa. COF-Rapa were dissolved in NS, and finally, 0.67 µg/µL COF nanoparticles could carry 300 µM rapamycin.

### 2.2. Transmission Electron Microscopy (TEM)

TEM samples were prepared as previously described by air-drying a drop of the nanocarriers solution on the surface of an ultrathin carbon film supported on copper grids [8]. TEM images were acquired on a JEM-2100 electron microscope (JEOL, Tokyo, Japan) at an accelerating 100–120 kV voltage.

### 2.3. Cell Culture and Cell Viability

BV2, Neuro-2a (N2A) cell lines were cultured in Dulbecco’s Modified Eagle’s Medium (DMEM; Gibco, USA) containing 10% (*v*/*v*) fetal bovine serum (FBS) (Gibco, Waltham, MA, USA) and 100U penicillin/100 g streptomycin (Sigma-Aldrich, St. Louis, MO, USA). Procell Life Science & Technology Co., Ltd. provided BV2 (Procell CL-0493). Shanghai Zhongqiaoxinzhou Biotech supplied N2A (ZQ0207). The cells were grown at 37 °C in a humidified 5% CO_2_ atmosphere. Cell viability was assessed by the Cell Counting Kit-8 assay (CCK-8, Dojindo, Japan). Cells were seeded in a 96-well plate with 8000 cells/well, and treated with varying concentrations of COF (0–1.34 µg/mL) for 6 or 24 h. Then, 10 μL CCK-8 solution was well added to each and incubated, at 37 °C, for 1.5 h. An iMark microplate reader (Victor3 1420 Multilabel Counter, Perkin Elmer, Waltham, MA, USA) was used to measure the absorbance at a wavelength of 450 nm.

BV2 cells were divided into three groups to test whether COF or rapamycin could inhibit microglial activation. After pre-treatment with COF (0.67 µg/mL) or rapamycin (300 nM) (1 × PBS and 1.8 × 10^−4^ % DMSO as control separately) for 2 h, LPS (1 μg/mL; Sigma-Aldrich, USA) and IFN-γ (100 ng/mL; Beyotime, Shanghai, China) were added for 18 h. Real-time quantitative PCR was used to test transcripts of pro-inflammatory factors.

### 2.4. Animals

Adult male C57BL/6J mice (6 weeks old, weighing 20 ± 2 g) were purchased from the model animal research center of Tongji Hospital. Mice were raised in a 12-h light/dark cycle environment, with free access to water at the model animal research center of Tongji Hospital. Animals were cared for and handled based on the ARVO Statement for the Use of Animals in Vision and Ophthalmic Research and the Ethics Committee of Tongji Hospital of Huazhong University of Science and Technology.

### 2.5. Acute Ocular Hypertension (AOH) Glaucomatous Model

In this model, transient acute elevation of IOP leads to retinal ischemia; thus, it is also referred to as the ocular ischemia–reperfusion (I/R) model. The retinal I/R model was established as described [16,17]. In brief, mice were anesthetized using an i.p. injection of 5% chloral hydrate. The pupils were dilated with 1% tropicamide. The corneas and conjunctivas were topically anesthetized with 2% lidocaine and cleaned with 0.5% iodophor. The anterior chamber of the right eye was cannulated with a 30-gauge infusion needle linked to a normal saline reservoir which was increased to 75 mmHg for 50 min [18,19]. The contralateral left eyes acted as controls. All experiments were performed at least three separate times.

### 2.6. Intravitreal Administration

Intravitreal administration was established as described previously [20,21]. In brief, after anesthesia and pupil dilation, a sclerotomy was created using a 30-gauge needle tip at 2 mm posterior to the limbus. Then, 1.5 µL of normal saline (NS), COF (0.67 ng/µL), rapamycin (300 µM) or COF-Rapa (0.67 ng/µL-300 µM) was injected into the vitreous body by using a 5-µL micro-syringe fitted with a 35-gauge needle (Hamilton, Reno, NV, USA). Injections were slowly delivered over 10 s with the needle left for 60 s to allow the diffusion of the solution and equilibration of IOP before removing the needle. Mice were then raised for 1–6 weeks before I/R injury or tissue collection.

### 2.7. RGC Labelling and Quantification

After the mice were anesthetized and their scalps shaved, they were placed in a stereotactic device (RWD Life Science, Shenzhen, China). The stereotaxic coordinates of the superior colliculi were set as 1.0 mm lateral and 1.0 mm anterior to lambda, at a depth of 1.6 mm. Fluoro-Gold (FG) (1 μL per injection of 4% (*w*/*v*) (Fluorochrome, LLC, Denver, CO, USA) was injected into both superior colliculi. It took nine days for retrograde transport of FG before retina collection to ensure proper RGC labelling [22]. After retinal flat mounts were prepared, FG-positive RGCs were identified using a fluorescent microscope (Olympus, BX51Olympus Co. LTD, Tokyo, Japan). Three continuous images of the optic nerve edge were collected using 40× objective lenses in two random quadrants. Surviving RGCs (gold dots) were counted. At least six retinas from different animals were used for each group.

### 2.8. Electroretinogram (ERG)

After pre-treatment with NS or COF by vitreous injection, changes in retinal function were assessed with scotopic ERGs one and four weeks after I/R injury. The mice were anesthetized under red light after 24 h of dark adaptation. The pupils were dilated with 1% tropicamide. Reference electrodes were connected to the corneas. Scotopic ERGs were recorded using a Diagnosys Celeris device (Diagnosys LLC 175 Cabot St., Beverly, MA, USA). Scotopic responses were elicited with eight flash intensities 10 cd × s/m^2^. The peak voltage of the a-waves (first negative ERG component) and b-waves (first positive ERG component) were automatically recorded.

### 2.9. Immunofluorescence and Immunohistochemistry

Mice were sacrificed, and eyes were fixed in 4% paraformaldehyde at the designated time points. Sections (5 μm) through the optic disk of the eye were treated with 10% donkey serum albumin for 60 min at room temperature and incubated, at 4 °C, overnight with combinations of primary antibodies (Table 1). For immunofluorescence, the sections were incubated with Dapi and species-specific fluorescently labelled immunoglobulin G secondary antibodies (donkey anti-rabbit Alexa Fluor 594, Thermo, Waltham, MA, USA, 1:400) in PBS at room temperature for 2 h. Then, the sections were examined using a fluorescent microscope (Bx51, Olympus Co., Tokyo, Japan). In the case of immunohistochemistry, the sections were incubated with 3,3′-diaminobenzidine and observed with a light microscope (Bx53, Olympus Co., Japan).

### 2.10. Real-Time Quantitative PCR (qPCR)

The primers (5′-3′) for the target mRNAs were as follows: Interleukin-1β (IL-1β): forward TGCCACCTTTTGACAGTGATG, reverse CCCAGGTCAAAGGTTTGGAA; inducible nitric oxide synthase (iNOS): forward GCTTGTCTCTGGGTCCTCTG, reverse CTCACTGGGACAGCACAGAA; tumor necrosis factor-α (TNF-α): forward ACGGCATGGATCTCAAAGAC, reverse AGATAGCAAATCGGCTGACG; β-actin: forward ATCTTCCGCCTTAATACT, reverse GCCTTCATACATCAAGTT. Total RNA of BV2 cells and retinas (48 h after I/R injury) were extracted with Trizol reagent (Takara Biomedical Technology, Beijing, China). The cDNAs were synthesized using PrimeScript RT Master Mix (TaKaRa). Quantitative analysis was performed by real-time qPCR using SYBR Advantage qPCR Premix Master Mix (TaKaRa) based on the standard protocol (Roche, Basel, Switzerland.). The amount of mRNA in the samples was measured, and the comparative Ct method was used to calculate the fold change in gene expression.

### 2.11. Western Blotting

Total protein was isolated from retinal samples by RIPA lysis solution (Beyotime, China) at the designated time points after I/R injury. Samples were separated with 12% polyacrylamide gels and transferred to polyvinylidene difluoride (PVDF) membranes. PVDF membranes were then blocked with 5% skimmed milk for 2 h, at room temperature, and incubated overnight, at 4 °C, with primary antibodies (Table 1) diluted in fetal bovine serum blocking solution. Subsequently, the PVDF membranes were incubated with species-specific horseradish peroxidase-conjugated secondary antibodies for one hour, at room temperature. Chemiluminescence substrate kits (ECL Plus; PerkinElmer Inc., Covina, CA, USA) were used to visualize protein expression levels. Target proteins were quantified and normalized to GAPDH using the Image-Pro Plus software 6.0 (version 6.0, Media Cybernetics). Relative changes in target proteins were calculated compared with the control and expressed as the “x fold change”.

### 2.12. Statistical Analysis

Data were presented as mean ± standard deviation (SD). One-way analysis of variance (ANOVA), the multiple comparisons test of with Dunnett and paired/unpaired *t*-tests (two-tailed) were performed using GraphPad Prism (version 8.0, GraphPad Software). *p* values less than 0.05 were considered statistically significant.

## 3. Results 

### 3.1. Synthesis and Characterization of COF Nanoparticles

COF-based nanoparticles were prepared in the same manner as previously described steps and methods in Figure 1A, similar to previous study [8]. The entire structure of nanocomposites can be thought of as a micelle with APTES-COF-1 acting as the oil phase and PEG-CCM acting as the surfactant. The nanoparticles dissolve in water and form a yellow suspension (Figure 1B). With a fluorescence microscope, the suspension showed fluorescence spots under the green channel excited at 488 nm and collected between 510 and 550 nm (Figure 1C). Transmission electron microscopy of the COF nanoparticles represented showed a scattered structure, with width ranging from 120 to150 nM (Figure 1D), similar to the previously described structures [8]. In addition, as previously stated, the nanoparticles were extremely stable in water and phosphate-buffered saline buffer (pH7.4) [8].

### 3.2. The Safety and Effect of COF Delivered In Vitro and In Vivo

We used CCK8 to test the effect of COF on BV2 and N2A to identify the influence of COF on cell viability, representing microglia cells and neuron cells in a mouse retina, respectively. Figure 2A shows that the cell viability of BV2/N2A cells treated with COF for 6 or 24 h exhibit no significant differences compared with control groups. COF was found to have anti-inflammatory properties by downregulating *IL-1β, TNF-α* and *iNOS* in BV2 cells using quantitative PCR. (Figure 2B) In in vivo experiments, the retinas were collected at one to six weeks after injection of COF into the vitreous cavity. Western blotting of retinas showed that the ratio of BCL2/BAX (a maker for anti/pro-apoptosis) [23] increased significantly at one week after COF injection. (Figure 2C) The number of RBPMS-positive RGCs (within 1 mm from the optic nerve) was not infected at three weeks and six weeks after COF injection, as confirmed by one-way ANOVA (*n* = 4, *p* < 0.05). (Figure 2D) Furthermore, anti-GFAP (maker for astrocytes) and anti-Iba1 (maker for microglia) showed no discernible differences 1–6 weeks after COF injection. (Figure 2E) The findings indicate that COF may be a safe drug or carrier in vitro and in vivo.

### 3.3. COF Could Efficiently Attenuate RGC Death, Increase Phototransduction Function and Alleviate the Overactivation of Microglia after Retinal I/R Injury

We aimed to investigate whether COF could protect RGCs from retinal I/R injury. Mice were randomly divided into two groups, whose both eyes were injected with 1.5 µL NS and COF, respectively. One week later, FG was injected into both superior colliculi of the mice. After three days, the right eyes of the mice were subjected to I/R injury, while the left eyes served as controls. Then, the eyes were collected and fixed in 4% paraformaldehyde (Figure 3A). Retinal flat mounts of FG-positive RGCs revealed that the survival rate of RGCs in the acute ocular hypertension group was significantly lower than in the left control group (0.999 ± 0.074 vs. 0.494 ± 0.112, *n* = 6, *p* < 0.01, Figure 3B,C). COF application may improve the RGC survival rate after I/R injury when compared to the NS control (0.698 ± 0.060, *n* = 9, *p* < 0.01; 0.777 ± 0.089, *n* = 12, *p* < 0.01).

To investigate the alternations in the retina after I/R injury and explore the mechanism of neuroprotective effect of COF nanoparticles, we performed RNA-seq analysis of retinas from control mice and I/R mice with or without COF treatments (Supported by Novogene Co., Ltd., Beijing, China). We analyzed differentially expressed genes, and found GO and KEGG enrichment. As is shown in Figure 3D, the top 20 GO terms were visualized in the bubble diagram, and the most enriched pathways were related to neuromuscular processes, axonogenesis and cell morphogenesis, among other things. The top 20 KEGG pathways enriched are visualized in Figure 3E. It suggests that COF may play a significant role in energy metabolism, apoptosis (p53 and HIF1-α signaling) and phototransduction.

We assessed the phototransduction function using the scotopic flash ERG, a massed rod photoreceptor response and rod-driven post-receptor neural retinal function [8]. The main components of the dark-adapted ERG are the cornea negative a-wave generated by the rod photoreceptors and the cornea-positive b-wave generated by the post-receptor activity [24]. The right eye underwent the acute glaucoma model, and the left ones served as non-I/R control after pre-treatment of vitreous injection of 1.5 µL NS or COF in both eyes for one week. Scotopic ERGs were examined one week (Figure 4A,B) after the I/R injury, and reductions in a wave and b wave were calculated between I/R and their controls. It shows that a-wave and b-wave responses are diminished in all groups in I/R eyes compared to none-I/R ones. The wave reductions were significantly smaller in the COF group, compared with the NS group at one week (Figure 4B).

Increasing evidence suggested that overactivated microglia may play pivotal roles in triggering neurotoxicity in the retina by producing pro-inflammatory factors [16,25,26]. We wanted to determine whether COF could play a role in inhibiting microglia overactivation in vivo. The representative retinal immunofluorescence image demonstrated an apparent decrease in iba1-labelled activated microglia following COF pre-treatment (Figure 4C,D). These findings demonstrated that COF may efficiently minimize the declines of the a-wave and b-wave in the scotopic responses, downregulate excessively activated microglia, and ultimately restore the function of RGCs.

### 3.4. Construction of COF-Rapamycin Nanoparticles: Single Intravitreal Injection, Long-Lasting Effect

APTES-COF-1 was synthesized, and APTES-COF-1@Rapamycin/Null and PEG-CCM were assembled (abbreviated as COF-Rapa and COF, respectively, encapsulated with rapamycin or not) (Figure 5A). Inside the cell, PEG-CCM is unplugged, and rapamycin is released by COF-Rapa due to the pH (acid)-triggered disintegration of COF, for example, in the case of nanoparticles being endocytosed by cells and shuttled to acidic compartments [8]. Thus, rapamycin was gradually released when the COF-Rapa were injected into the vitreous cavity.

In the present study, phosphorylated S6 ribosomal protein (pS6), a downstream marker of mTORC1 pathway activity, was used to detect the effect of released rapamycin [10,27]. We delivered NS (control), COF, COF-Rapa, or rapamycin into mice eyes separately to test whether they could affect the activity of the mTORC1 pathway for a few days after the intravitreal injection. One week later, retinas were collected, and the Western blotting analysis indicated that only COF-Rapa could downregulate pS6 significantly compared with control (*n* = 3, *p* = 0.004) (Figure 5B,C). A single dose of rapamycin could not inhibit the activity of mTORC1 for even one week (*n* = 3, *p* = 0.202) (Figure 5B,C). In addition, COF showed a negligible effect on mTORC1 activity compared with normal saline (*n* = 3, *p* = 0.073). (Figure 5B,C)

The retinas were collected one to six weeks after injection to ascertain the duration of rapamycin releasing from the COF-Rapa system in the vitreous cavity. Western blotting showed that pS6 was inhibited within six weeks after the COF-Rapa injection by one-way ANOVA (*n* = 4, *p* < 0.05). (Figure 2D,E) Immunohistochemistry also revealed that pS6 was downregulated at different layers of the mice retinas within six weeks after the intravitreal injection. (Figure 2F) Therefore, rapamycin from a single injection might not be effective for very long, while the COF-Rapa system might work for at least six weeks.

### 3.5. COF-Rapa Could Efficiently Alleviate the Inflammatory Reaction in Retinal I/R Injury

As before, mice were randomly divided into four groups, and both eyes were delivered with 1.5 µL NS, COF, COF-Rapa (300 µM), or rapamycin (300 µM). After one week, both eyes experienced I/R injury. Two days later, the mice were killed, and the retinas were used to extract RNA or proteins. As is shown, transcripts of pro-inflammatory factors, including IL-1β, TNF-α, and iNOS, were reduced in the eyes after I/R injury with COF-Rapa and COF injection, compared to NS control ones. (Figure 6A)

TLR2/4 was reported to be the upstream genes triggering pyrosis and apoptosis, and nucleotide-binding oligomerization domain-like receptor family caspase-activation and recruitment domain containing 3 (NLRP3) is the principal inflammasomes involved in ocular I/R injury [16,19,28]. The Western blotting showed that TLR2/4 and NLRP3 were downregulated in COF-Rapa-treated eyes, accompanied by the inhibition of pS6 compared with other groups (Figure 6B). These results reflect that rapamycin, which was delivered by COF, inhibited inflammatory reactions by downregulating TLR2/4-NLRP3 pathways and lowering pro-inflammatory factors, hence preventing the loss of RGCs (Figure 6C).

## 4. Discussion

In our study, a water-dispersible COF-based nanoplatform was created. The COF nanoparticles appear to be a good drug itself because they can improve retina function and reduce microglia overactivity, finally effectively attenuate RGC death following retinal I/R injury. Additionally, COF might serve as a carrier for medication release to protect RGC in glaucoma models. 

The rapamycin-loaded, COF-based nanoparticles appear to have a number of benefits when used locally to treat glaucoma and other ocular ailments. First, the COF-Rapa and COF nanoparticles were highly dispersed and stable in water with a pH of 7.4 and could be stored at room temperature [8]. Second, the release of encapsulated rapamycin depends on the pH-triggered disintegration of COF in the acidic environment, and this characteristic determines that it can play a role both in vitro and in vivo [8]. Notably, COF-Rapa can take effect in two different ways when injected into the vitreous cavity. On the one hand, the cells endocytosed nanoparticles and could transport them into acidic compartments. By releasing pro-inflammatory substances, microglia, which have a high phagocytic function in the retina, play a critical role in causing neurotoxicity [16,25,26]. Thus, we tested the effect of our nanoparticles on inhibiting microglia overactivation following retinal ischemia/reperfusion injury. The findings imply that microglial endocytosis of nanoparticles may advantageously allow for attenuation of retinal function by lowering pro-inflammatory factors. On the other hand, disorders or injuries, such as diabetes, I/R and retinal arterial occlusion, can result in acidosis in the retina [29,30,31]. Increased H^+^ in the tissue may also cause disintegration of COF and rapamycin release, too. So, rapamycin may possibly directly affect ganglion cells in addition to microglia. Third, our findings indicate that COF-loaded medications can have a long-lasting effect with a single intravitreal injection in vivo, avoiding the need for a high dose of systemic treatment for eye disease and lowering the risk of consequences from repeated intravitreal injection.

Last but not least, rapamycin, a potential medicine for RGC protection, can produce a more noticeable effect when delivered through COF carrier than when it is in pure lipophilic form. In addition to suppressing overactive microglia, the released rapamycin may directly affect RGCs and lessen their damage through autophagy and glucose metabolism, as reported before [12,13]. Rapamycin may also inhibit the degradation of sirtuin1, which is an NAD^+^-dependent deacetylase important for stress responses and cell survival [32,33,34]. Additionally, the MAPK pathway could be affected by mTORC1 activity [35,36] and might be connected to RGC survival by COF nanoparticles in our glaucoma model.

Our study also has some shortcomings. The IOP induced here is higher than that measured in most patients with glaucoma, and the anatomy of the mouse optic nerve is different from that of humans [37,38]. Whether these results apply to glaucoma in human beings remains unclear. In addition, even though the sustained-release duration of COF-Rapa was measured, more specific pharmacodynamics needed to be quantified. More trials are required to test the safety of COF for different tissues in the eye.

## 5. Conclusions

A water-dispersible COF-based nanoplatform was constructed for the release of rapamycin. With just one intravitreal injection, it was successful in attenuating RGC death and preserving retinal function for an extended period of time. This research may lead to the development of novel local glaucoma therapies and inspire the fabrication of multifunctional COF-based biomaterials for ocular disorders.

## Figures and Tables

**Figure 1 polymers-14-03265-f001:**
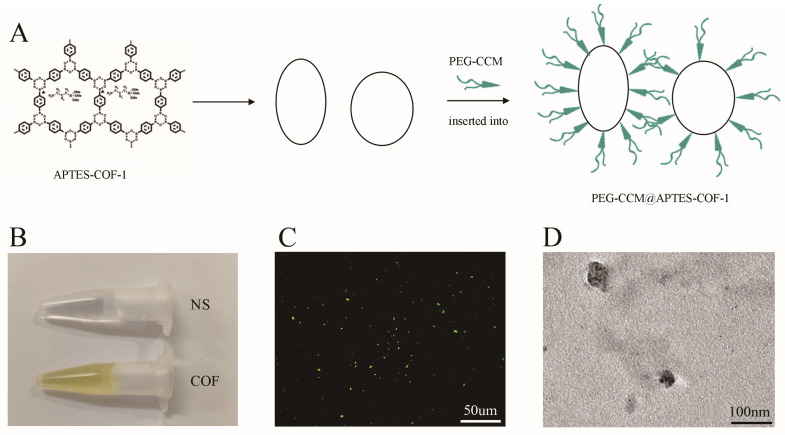
Schematic representation of COF and its characteristics. (**A**): Schematic structures and synthesis of PEG-CCM@APTES-COF-1 (abbreviated as COF). (**B**): Digital photographs of dispersions of COF; (**C**): Representative green channel image recorded for COF dispersion; (**D**): Transmission electron microscope of COF nanoparticles.

**Figure 2 polymers-14-03265-f002:**
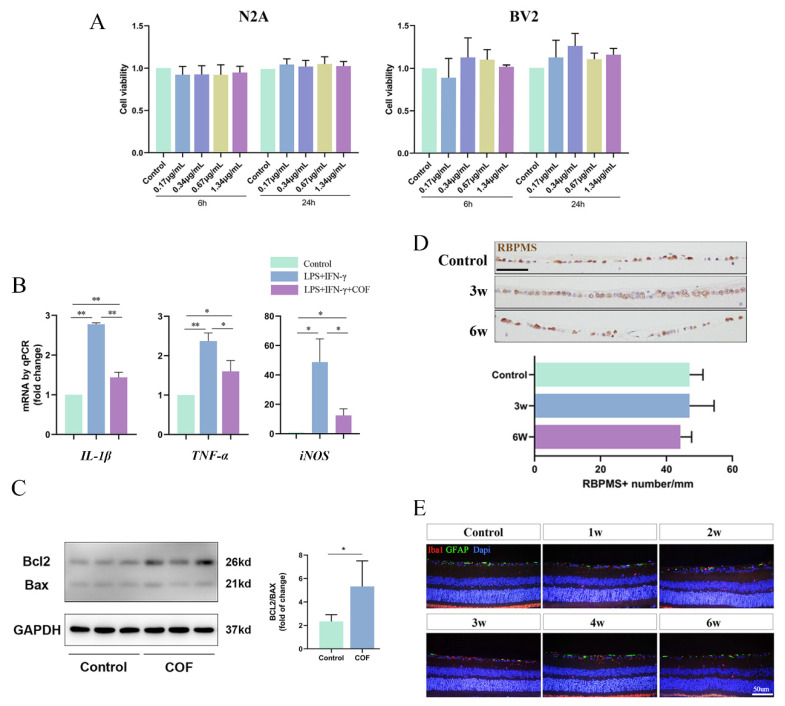
The effect of COF delivered in vitro and in vivo. (**A**): The viabilities of N2A and BV2 cells, co-cultured with COF (0–1.34 μg/mL) for 6 and 24 h, were measured by CCK-8 assay kit (*n* = 6). (**B**): Quantitative PCR of *IL-1β, TNF-α* and *iNOS* in BV2 cells, pretreated with normal saline or COF (0.67 μg/mL), following stimulated by LPS (1 μg/mL) and IFN-γ (100 ng/mL) for 18 h (*n* = 3). (**C**): Western blotting and statistical graph of BCL2/BAX of retinas after intravitreal injection normal saline and COF. (**D**): Representative immunohistochemical images and statistical graphs of the number of RBPMS-positive RGCs in retinas three and six weeks after single intravitreal injection COF. (*n* = 4). Scale bar = 50 um. * indicates *p* < 0.05, ** indicates *p* < 0.01. (**E**): Representative immunohistochemical images of anti-GFAP and anti-Iba1 in retinas one to six weeks after single intravitreal injection COF. (*n* = 4).

**Figure 3 polymers-14-03265-f003:**
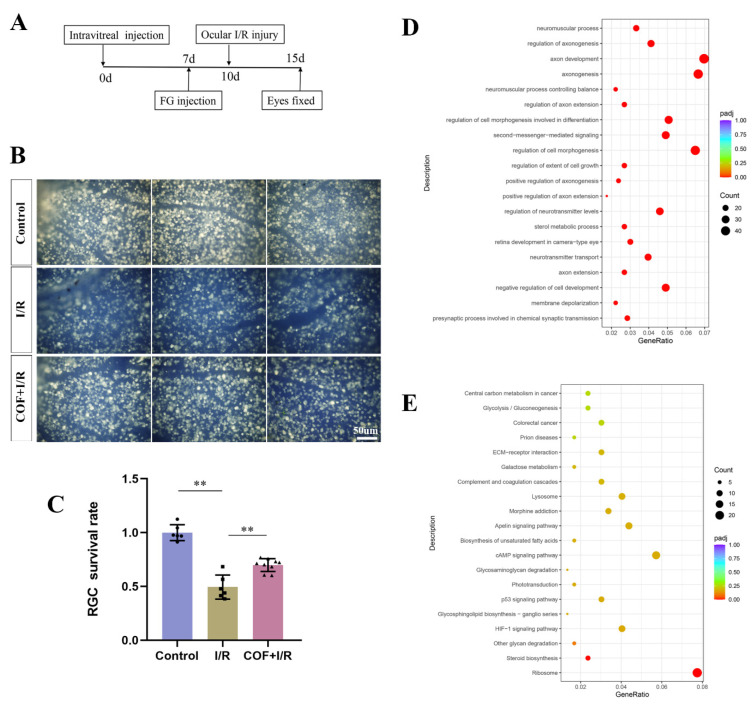
COF could efficiently attenuate acute elevated-IOP-induced RGC death. (**A**): Schematic representation of treatments before retina patching. (**B**): Representative continuous images (40×) of Fluoro-Gold labelled RGCs in entire mount retinas. (**C**): Quantitative analysis showed that the number of surviving RGCs increased significantly in COF injected ischemic eyes compared to NS control (*n* = 6–12). (**D**,**E**): GO ad KEGG pathway enrichment analysis in ischemic retinas 5 days after reperfusion between COF group and NS control. ** indicates *p* < 0.01.

**Figure 4 polymers-14-03265-f004:**
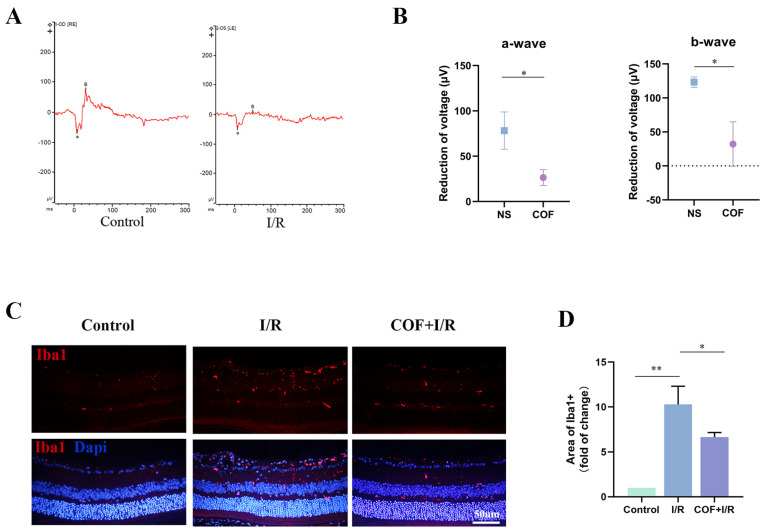
COF could efficiently increase phototransduction function and attenuate microglia activation after I/R injury. (**A**): Representative traces by scotopic flash ERG one week after ocular I/R or without. The eyes of mice received an intravitreal injection of 1.5 ul NS or COF one week before the I/R model. Then, the right eye underwent the ocular I/R injury, and the left served as control. (**B**): The reduction in a/b-wave voltage = a/b-wave of control minus a/b-wave of I/R eye. (*n* = 5). (**C**,**D**): Representative immunofluorescence image and quantitative analysis of Iba1+ microglia in control, ischemic retinas and retinas with COF pre-treated 72 h after reperfusion. (*n* = 3–5). * indicates *p* < 0.05. ** indicates *p* < 0.01.

**Figure 5 polymers-14-03265-f005:**
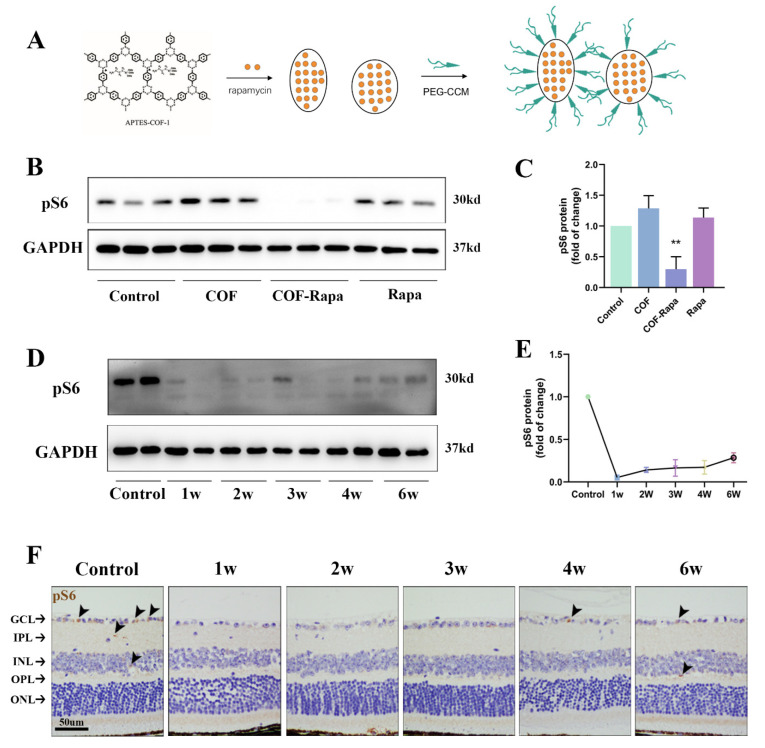
Duration of sustained release of rapamycin by COF with single intravitreal injection. (**A**): Schematic structures and synthesis of PEG-CCM@APTES-COF-1@Rapamycin (abbreviated as COF-Rapa). (**B**,**C**): Western blotting images of retinas one week after intravitreal injection of NS (control), COF, COF-Rapa or rapamycin. Phosphorylated S6 ribosomal protein (pS6) was significantly inhibited in the COF-Rapa group (*n* = 3). (**D**,**E**): Representative Western blotting of retinas at one-to-six weeks after single intravitreal injection COF-Rapa. The pS6 activity was significantly inhibited for at least six weeks. (**F**): Representative immunohistochemical images of retinas at one-to-six weeks after single intravitreal injection COF-Rapa. The arrowhead points to the pS6-positive cells. GCL, ganglion cell layer; ICL, inner plexiform layer; INL, inner nuclear layer; OPL, outer plexiform layer; ONL, outer nuclear layer. Scale bar = 50 µm. ** indicates *p* < 0.01.

**Figure 6 polymers-14-03265-f006:**
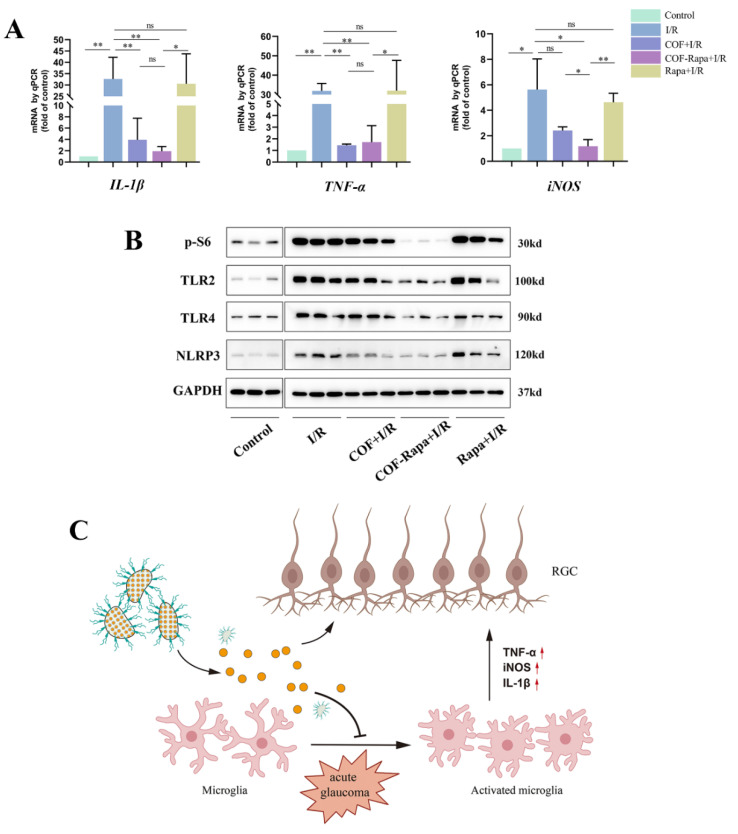
COF-Rapa could exert stronger anti- inflammatory effect than COF in retinal I/R injury. (**A**): Quantitative PCR of *IL-1β*, *TNF-α* and *iNOS* in control and ischemic retinas 48 h after reperfusion, pre-treated with 1.5 µL NS, COF, COF-Rapa or rapamycin. (*n* = 3). (**B**): Western blotting of control and ischemic retinas 72 h after reperfusion, indicating decreased expression of pS6, TLR2/4 and NLRP3 in COF-Rapa-injected eyes (*n* = 3). (**C**): Schematic representation of the effect of COF-Rapa in attenuating RGCs injury in the mouse model of acute glaucoma. * indicates *p* < 0.05, ** indicates *p* < 0.01, ns = not significant.

**Table 1 polymers-14-03265-t001:** Primary antibodies used in this study.

Antigen	Species	Supplier	Catalog Number	Dilution (WB)	Dilution (IF/IHC)
Bcl2	Rabbit	Proteintech	26593-1-AP	1:1000	-
BAX	Rabbit	Proteintech	50599-2-Ig	1:1000	-
pS6	Rabbit	CST	5364	1:1000	1:500
RBPMS	Rabbit	Abcam	ab194213	-	1:800
caspase3	Rabbit	CST	9662	1:1000	-
Iba1	Rabbit	Abcam	Ab178846	-	1:800
TLR2	Rabbit	Abcam	Ab209217	1:1000	-
TLR4	Rabbit	Sevicebio	GB11519	1:1000	-
NLRP3	Rabbit	Boster	BM4490	1:200	-
GAPDH	Mouse	Proteintech	600004-1-Ig	1:10,000	-

Bcl2: B-cell lymphoma-2; pS6: Phospho-S6 ribosomal protein; TLR2/4: Toll-like receptor 2/4; Iba1: Ionized calcium binding adapter molecule 1; RBPMS: RNA Binding Protein; TLR4: Toll-like receptor 4; NLRP3, Nucleotide-binding oligomerization domain-like receptor family caspase-activation and recruitment domain containing 3; GAPDH: Glyceraldehyde-3-phosphate dehydrogenase.

## Data Availability

The raw data required to reproduce these findings will be made available upon request.

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
