# Peer review of "Covalent Organic Framework (COF): A Drug and Carrier to Attenuate Retinal Ganglion Cells Death in an Acute Glaucoma Mouse Model"

_polymers, 2022, doi:10.3390/polym14163265_

Round 1
Reviewer 1 Report
1. The experimental mouse model used in this study is questionable. The IOP elevated in the hypertensive animals were ~75mmHg, which is way higher than any known glaucoma pathological conditions in mouse or in human. Though the authors have identified this issue as a shortcoming of the present study it does not necessarily indicate any useful information regarding our understanding in the hypertensive glaucoma pathology. How was the control/ contralateral eyes prepared? Do they also have similar cannulation procedure performed on them with NS perfusion at physiological pressure (~15mmHg)?
2. The retinal preparation does not specifically include any identifying or characterizing steps of the cells present in the prep and at what percentage. Elevated IOP-induced astro-glial altered protein expression has been a subject of study for long past and that alone never conclusively indicated any plausible mechanisms uniformly applicable for hypertensive and normotensive glaucoma. However, most of the recent research has been directed to understand the cause of axonal cell death in RGC and efforts are being made for axonal regeneration. In this regard, the present research does not clearly indicate anything about the target cell type(s) and possible insights regarding explanation of COF-treatment induced protection in the mouse RGC.
3. All WB data presented in the manuscript do not conclusively establish the identity of the target bands. Please provide with images showing adjacently ran mol wt standards. Why 12% SDS PAGE have been used to resolve TLR2 (11kDa) and NLRP3 (120kDa). For proteins >100kda mol wt, 7.5% SDS PAGE would be a better choice. Please provide the validation information for each primary antibodies used against mouse targets.
4. PCR etc: The quality of primer design was not adequately controlled. A quick BLAST analysis did not pick always the same accession ID. This discrepancy is very prominent in the TNFα forward primer selection as the target hit falls way below the initial 20 hits and mostly picked mouse genome assembly, chromosome: 17. Please provide the accession number of each target gene and also the amplicon size.
5. The discussion section of the manuscript is underdeveloped and to be rewritten discussing the present experimental finding and comparing with the existing knowledge in the specific field.
Minor:
1. Correct all ‘ug’ and ‘ul’ to ‘µg’ and ‘µl’
2. Figures 3C and 3E were wrongly cited on page 6.
3. Citation of figure 5A is missing on page 10
Author Response
Dear reviewer,
We appreciate your insightful comments on our manuscript. The recommendations have been carefully reviewed, and some adjustments have been made. We made changes to the manuscript and did our best to make it better. Revisions were highlighted in red. Point-by-point revision notes are provided as follows:
- The experimental mouse model used in this study is questionable. The IOP elevated in the hypertensive animals were ~75mmHg, which is way higher than any known glaucoma pathological conditions in mouse or in human. Though the authors have identified this issue as a shortcoming of the present study it does not necessarily indicate any useful information regarding our understanding in the hypertensive glaucoma pathology. How was the control/ contralateral eyes prepared? Do they also have similar cannulation procedure performed on them with NS perfusion at physiological pressure (~15mmHg)?
Response:
Thank you for your suggestions. We decide to apply the I/R model for several reasons. First, there is a rapid rise in IOP to levels above retinal perfusion pressure in patients with acute primary angle closure glaucoma (a kind of glaucoma in which the IOP could be 60mmHg), leading to retinal ischemia and RGC mortality. This is consistent with the I/R model. The second is that, in accordance with research that have been published (①②), acute ocular hypertension (AOH) glaucomatous model is just utilized to mimic the transient acute elevation of IOP which leads to retinal ischemia; Thus, it is also referred to as the ocular ischemia-reperfusion (I/R) model. Additionally, the AOH model is stable, dependable, and affordable when compared to alternative glaucomatous models (such as magnetic beads injection into the anterior chamber, cauterization of superior scleral, etc.). Thus, we used the AOH model in this study to mimic the retinal ischemia and reperfusion injury in glaucoma retinas. In our study, the control involved a sham operation that didn't raise the pressure in the left eye on the opposite side.
①Caspase-8 promotes NLRP1/NLRP3 inflammasome activation and IL-1β production in acute glaucoma. Proc Natl Acad Sci U S A. 2014
②P16INK4a upregulation mediated by TBK1 induces retinal ganglion cell senescence in ischemic injury[J]. Cell Death & Disease, 2017
- The retinal preparation does not specifically include any identifying or characterizing steps of the cells present in the prep and at what percentage. Elevated IOP-induced astro-glial altered protein expression has been a subject of study for long past and that alone never conclusively indicated any plausible mechanisms uniformly applicable for hypertensive and normotensive glaucoma. However, most of the recent research has been directed to understand the cause of axonal cell death in RGC and efforts are being made for axonal regeneration. In this regard, the present research does not clearly indicate anything about the target cell type(s) and possible insights regarding explanation of COF-treatment induced protection in the mouse RGC.
Response:
Thanks for your comment. Under a surgical operation microscope, the entire retina was separated from the eyeball for our investigation. The normal procedure was followed to extract the total RNA from the retinas using the Trizol reagent (Takara Biomedical Technology, China). Retinal proteins were obtained based on the RIPA lysis solution protocol (P0013B; Beyotime, China). This information has been added to the Methods part.
In the current work, we primarily focus on microglia (for example, the reduction of the maker Iba1, NLRP3, etc.). We believe COF could lessen the inflammatory reaction and subsequent RGC loss. We did not consider whether COF could promote axon regeneration, though. This is a weakness in our findings and warrants more investigation.
- All WB data presented in the manuscript do not conclusively establish the identity of the target bands. Please provide with images showing adjacently ran mol wt standards. Why 12% SDS PAGE have been used to resolve TLR2 (11kDa) and NLRP3 (120kDa). For proteins >100kda mol wt, 7.5% SDS PAGE would be a better choice. Please provide the validation information for each primary antibodies used against mouse targets.
Response:
I appreciate your advice. In WB tests, we discriminated where the target band was by referring to the maker and looking at WB images in published literature. Through many trials, 12% SDS PAGE can display 100-120kd protein well by appropriately extending the electrophoresis time, and this can also retain the target band of GAPDH. Each primary antibody used in this study has been proved to be able to react with mouse species in the instructions which all included a lot of cited sources. Websites with the links below can be accessed for this information.
Anti-Bcl2 antibody,https://www.ptgcn.com/products/Bcl2-Antibody-26593-1-AP.htm
Anti-Bax antibody, https://www.ptgcn.com/products/BAX-Antibody-50599-2-Ig.htm
Anti-GAPDH, https://www.ptgcn.com/products/GAPDH-Antibody-60004-1-Ig.htm
Anti-pS6, https://www.cellsignal.cn/products/primary-antibodies/phospho-s6-ribosomal-protein-ser240-244-d68f8-xp-rabbit-mab/5364?site-search-type=Products&N=4294956287&Ntt=5364&fromPage=plp&_requestid=860915
Anti-caspase3, https://www.cellsignal.cn/products/primary-antibodies/caspase-3-antibody/9662?site-search-type=Products&N=4294956287&Ntt=9662&fromPage=plp
Anti-NLRP3, https://www.boster.com.cn/home/product/anti-nlrp3-antibody_bm4490.html
Anti-TLR2, https://www.abcam.cn/tlr2-antibody-epr20303-ab209217.html
Anti-TLR4, https://www.servicebio.cn/goodsdetail?id=1260
- PCR etc: The quality of primer design was not adequately controlled. A quick BLAST analysis did not pick always the same accession ID. This discrepancy is very prominent in the TNFα forward primer selection as the target hit falls way below the initial 20 hits and mostly picked mouse genome assembly, chromosome: 17. Please provide the accession number of each target gene and also the amplicon size.
Response: Thanks for your advice. We have been actively evaluating the primer.
- IL-1β: forward TGCCACCTTTTGACAGTGATG, reverseCCCAGGTCAAAGGTTTGGAA;
https://blast.ncbi.nlm.nih.gov/Blast.cgi
gene accession number: AL808143;amplicon size =91
- iNOS: forward GCTTGTCTCTGGGTCCTCTG, reverse CTCACTGGGACAGCACAGAA;
https://blast.ncbi.nlm.nih.gov/Blast.cgi
gene accession number:AF427516; amplicon size = 217
- TNF-α: forward ACGGCATGGATCTCAAAGAC, reverse AGATAGCAAATCGGCTGACG;
https://blast.ncbi.nlm.nih.gov/Blast.cgi
gene accession number:AB039224;amplicon size = 138
- β-actin: forward ATCTTCCGCCTTAATACT, reverse GCCTTCATACATCAAGTT.
https://blast.ncbi.nlm.nih.gov/Blast.cgi
gene accession number:AC144818;amplicon size = 102
We think the designed primers might be able to satisfy PCR's criteria after blasting for mouse species. To better complete PCR studies, we should undoubtedly put more effort into learning about primer design.
- The discussion section of the manuscript is underdeveloped and to be rewritten discussing the present experimental finding and comparing with the existing knowledge in the specific field.
Response:
Thanks for your advice. In this study, we mainly focused on the application of nanomaterials in ocular disease. So, in the discussion part we concentrated on the advantages and potential mechanisms by COF application in the glaucoma model.
Minor:
- Correct all ‘ug’ and ‘ul’ to ‘µg’ and ‘µl’
- Figures 3C and 3E were wrongly cited on page 6.
- Citation of figure 5A is missing on page 10
Response: Thank you very much for discovering these errors. We have corrected it and highlighted in Red.
Reviewer 2 Report
Reviewer report on manuscript Polymers-1800973
The submitted research presents data regarding to the application of COF as carrier to attenuate retinal ganglion cell death in acute glaucoma mouse.
After careful research of the electronic databases, I agree with the authors' claims about the originality of their method. However, there are some minor comments that need to be addressed prior to its final acceptance.
Comments
1) The manuscript should be revised linguistically.
2) Section 2.3 “cell culture and cell viability” should start as a new paragraph.
3) The unit ng/ul should be corrected as “ng/μL”
4) Section 2.8: correct as 10 cd x s/m2.
5) The conclusions of this research should be described in a separate section.
Author Response
Dear reviewer,
Thank you for your comments on our manuscript. We have sought help from a native speaker to abolish the manuscript. We have made changes according to your suggestions one by one. The revisions were highlighted in red in the new manuscript. Thanks again.
Reviewer 3 Report
Xi and Wang et al. in their paper entitled "Covalent organic framework (COF): a drug and carrier to attenuate retinal ganglion cells death in an acute glaucoma mouse model," (Manuscript Number: polymers-1800973) demonstrated that COF could be an RGC protection drug, and a carrier for rapamycin sustained release. The results of this study may be useful to the scientific community working on the aforementioned subject. Before this work may be published, the following issues must be resolved.
Comments:
1. The complete form of an abbreviation, such as RGC, must be used the first time (please check the abstract).
2. Section 2.3. Cell culture and cell viability should come as a new section.
3. Authors need to explain how APTES-COF-1 and PEG@CCM interact.
4. According to TEM, the particle size is in the nm range. The optimum method to support the formation of PEG-CCM@APTES-COF-1 will be DLS and zeta potential measurement. The author should include the DLS and Zeta potential measurement data.
5. The manuscript's English writing has to be improved.

Author Response
Dear reviewer,
Thank you for your comments on our manuscript. We have made some changes to the manuscript based on your suggestions. The revisions were highlighted it in red. The following are point-by-point revision notes.
- Thecomplete form of an abbreviation, such as RGC, must be used the first time (please check the abstract).
Response:We have checked the abstract and main text to ensure the complete form of an abbreviation when it was used at the first time. The modifications were highlighted it in red.
- Section 2.3. Cell culture and cell viability should come as a new section.
Response:This has been corrected.
- Authors need to explain how APTES-COF-1 and PEG@CCM interact.
Response:Thanks for your advice. This interaction is important. In detail, two milligrams of APTES-COF-1 and 2 mg of PEG-CCM were dispersed in 1 mL of 1,4-dioxane, stirred and maintained at a predetermined temperature with a water bath for 20 min. Afterwards, 2 mL of water was slowly added within 1 h, followed by the injection of 7 mL of water within 1 h. After stirring for another 2 h, residual PEG350-CCM and 1,4-dioxane were removed by dialysis (MWCO 1 kDa) against deionized water, after freeze-drying.
We only briefly described this process because it has previously been elaborated by Guiyang Zhang et al (1) and it is cited in the manuscript.
(1) Guiyang Zhang, et.al. Water-dispersible PEG-curcumin/amine-functionalized covalent organic framework nanocomposites as smart carriers for in vivo drug delivery. Nat Commun. 2018;9(1):2785.
- According to TEM, the particle size is in the nm range. The optimum method to support the formation of PEG-CCM@APTES-COF-1 will be DLS and zeta potential measurement. The author should include the DLS and Zeta potential measurement data.
Response:Your suggestion is very good. Because the structural characterizations of PEG-CCM@APTES-COF-1 have been reported, including TEM ((1), Fig.1) and DLS ((1), Supplementary Fig. 8), so we have chosen to present here a typical TEM image in this study.
(1)Guiyang Zhang, et.al. Water-dispersible PEG-curcumin/amine-functionalized covalent organic framework nanocomposites as smart carriers for in vivo drug delivery. Nat Commun. 2018;9(1):2785.
- The manuscript's English writing has to be improved.
Response:We have sought help from a native speaker to abolish the manuscript. Revisions were highlighted it in red. Thanks for your advice again.
Round 2
Reviewer 1 Report
The revised manuscript has been improved significantly. The authors have provided satisfactory answers to the comments I had.